# Fluorescent Biosensors for the Detection of Viruses Using Graphene and Two-Dimensional Carbon Nanomaterials

**DOI:** 10.3390/bios12070460

**Published:** 2022-06-27

**Authors:** Ahmed M. Salama, Ghulam Yasin, Mohammed Zourob, Jun Lu

**Affiliations:** 1State Key Laboratory of Chemical Resource Engineering, Beijing University of Chemical Technology, Beisanhuan East Road 15, Beijing 100029, China; 2019420035@mail.buct.edu.cn; 2Institute for Advanced Study, Shenzhen University, Shenzhen 518060, China; yasin@mail.buct.edu.cn; 3Department of Chemistry, Alfaisal University, Riyadh 11533, Saudi Arabia

**Keywords:** FRET sensing, 2D carbon material-based sensors, multiplexing virus detection, limit of detection, recognition element

## Abstract

Two-dimensional carbon nanomaterials have been commonly employed in the field of biosensors to improve their sensitivity/limits of detection and shorten the analysis time. These nanomaterials act as efficient transducers because of their unique characteristics, such as high surface area and optical, electrical, and magnetic properties, which in turn have been exploited to create simple, quick, and low-cost biosensing platforms. In this review, graphene and two-dimensional carbon material-based fluorescent biosensors are covered between 2010 and 2021, for the detection of different human viruses. This review specifically focuses on the new developments in graphene and two-dimensional carbon nanomaterials for fluorescent biosensing based on the Förster resonance energy transfer (FRET) mechanism. The high-efficiency quenching capability of graphene via the FRET mechanism enhances the fluorescent-based biosensors. The review provides a comprehensive reference for the different types of carbon nanomaterials employed for the detection of viruses such as Rotavirus, Ebola virus, Influenza virus H_3_N_2_, HIV, Hepatitis C virus (HCV), and Hepatitis B virus (HBV). This review covers the various multiplexing detection technologies as a new direction in the development of biosensing platforms for virus detection. At the end of the review, the different challenges in the use of fluorescent biosensors, as well as some insights into how to overcome them, are highlighted.

## 1. Introduction

There is competition in the development of graphene-based biosensors in the market [1]. A large number of research articles have been published, exceeding 3600 articles since 2010, focusing on fluorescent biosensors. Biosensors have enormous applications in various areas, including (a) diagnostic applications [2], bioprocess monitoring [3], and improving the quality of new pharmaceutics [4]; (b) environmental testing [5]; and (c) food quality [6].

Biosensors can be defined as miniaturized devices incorporating recognition elements for identifying and/or quantifying either a synthetic or biochemical analyte. The recognition element can be an antibody, whole cell, aptamer, peptides, or DNA [7,8,9]. Biosensors can provide real-time outputs with high sensitivity, high specificity, and a low limit of detection [10,11]. Optical biosensors are a major group of transducers with unique features in comparison with other biosensors [12,13]. The optical detection mechanism relies upon the interaction between a recognition element and the optical transducer [14]. Optical biosensing can be characterized into two general categories: label-based and label-free sensors [15,16].

In the label-free sensors, the detected signal is the result of the direct interaction of the analyte with the immobilized recognition element on the transducer’s surface [17,18]. On the other hand, the label-based sensors require the employment of labels, such as colorimetric, fluorescent, or luminescent, to produce the optical signal [19,20,21]. There are various types of optical biosensors reported in the literature [22]. This review will cover the fluorescent-based biosensors used for the detection or identification of viruses.

To the best of the authors’ knowledge, no review has been published so far covering the use of carbon nanomaterials for virus detection using fluorescence. As a consequence, this recent literature review reports the virus detection using fluorescent biosensors based on carbon nanomaterials [22]. Moreover, this review will focus on the recent improvements in graphene-based biosensors for human virus detection using multiplexed detection based on a FRET–graphene oxide biosensor [23].

## 2. Graphene Oxide Fluorescent Biosensor for Human Virus Detection

In 2004, Novoselov and Geim synthesized and characterized a single sheet of graphene with covalently linked atoms arranged in one or multiple atomic layer planes, which form bulk layered materials via weak van der Waals gaps [24,25,26]. Since then, the application of graphene has grown exponentially in several biomedical fields owing to its well-defined properties and high specific surface area [27]. Therefore, graphene research has progressed to the development of more adaptable and customizable 2D alternatives with a higher composition, structure, and diversity in functionality [28]. Graphene-derived materials, including graphene oxide (GO) and reduced graphene oxide (rGO), are extensively used in various biosensing devices and assays. In-vivo and in-vitro, the target biomolecule is fluorescently labeled, which is identified using fluorescence microscopy or fluorescence spectrometry [12].

Recently, graphene oxide (GO) has gained a great deal of interest for virus detection due to its large surface area, 2D carbon structure, electrical conductivity, thermal robustness, flexibility, optical transparency, and mechanical characteristics, with high chemical stability [29,30,31,32,33,34]. Aptamers have demonstrated that graphene possesses unique attachment, molecular recognition, and biocompatibility via linking them with graphene to enhance the sensitivity and selectivity of the manufactured biosensors. Antigen–antibody and aptamer–target interactions, essentially were previously widely employed to create biosensors for the detection of analytes. The aptamer–target system-based aptasensors have several superior features over antigen–antibody system-based biosensors, including higher affinity, reduced costs, simpler fabrication, higher sensitivity, and wide range of analyte-sensing applications [35].

### 2.1. Fluorescence Resonance Energy Transfer Mechanism (FRET)

FRET is an effective approach to the quantitative determination of biomolecules with high specificity and sensitivity [36]. A fluorophore probe is absorbed on the surface of a quencher (graphene) to constitute a FRET pair, as shown in Figure 1. Graphene-like 2D nanomaterial is a powerful fluorescence quencher, that increased the application of fluorescent sensors to achieve a sensitive detection platform, for various targets via an assembly or conjugation mechanism. The binding force between the fluorophore and target biomolecules decreases the fluorophore–graphene interaction in the presence of the target biomolecules, resulting in the fluorophore’s release from the graphene surface to restore the dye’s fluorescence [37].

GO has combined electron–hole pairs positioned inside sp^2^ carbon clusters, incorporated within the sp^3^ matrix, and exhibits UV to near-infrared (NIR) light absorption [38,39,40]. The large dislocations in the p-electrons of GO display an exceptional ability to quench the fluorescence emitted by fluorescent dyes or quantum dots. Therefore, GO can be considered a good option as a quencher in FRET-based biosensors, as it offers low background noise and a high signal-to-noise ratio [41].

### 2.2. Characteristics of Graphene Material and Biomolecule Interaction

There are several functional groups—for instance, carboxyl, epoxide, and hydroxyl groups—at the surface and edges of GO that can be used for the immobilization of recognition elements via covalent bonding, electrostatic interaction, and hydrogen interactions.

Due to the honeycomb structure of the graphene oxide lattice, inserting suitable functional groups enhances the optical properties. The chemically active, soluble, hybrid graphene surfaces result from the functionalization of pure graphene via covalent and noncovalent attachment [42,43].

Hydrogen bonding, van der Waals interactions, and π–π interactions are forms of non-covalent functionalization, whereas covalent bonding using 1-ethyl-3-(3-dimethylaminopropyl)carbodiimide hydrochloride/N-hydroxy sulfosuccinimide (EDC/NHS) chemistry and others has been utilized to attach a variety of recognition receptors onto the graphene biosensor surface (Figure 2) [44]. Fluorescent-labeled single-stranded DNA (ssDNA) was adsorbed onto the GO molecule surface due to π–π stacking [12,45,46,47]. Through van der Waals interaction, partially reduced GO(p-rGO) or rGO interacts with numerous biomolecules, such as proteins and DNA [46]. The functionalization of GO by inserting different functional groups with high oxygen content helped in developing a functional pathogen biosensor.

GO sheets have higher and more diverse adsorption capacities, which may increase the interaction with the pathogen and virus, which in turn enhances the sensing performance [48]. rGO lacks hydroxyl groups, so it has lower solubility in cell culture media, limiting its application to the evaluation of cellular biomaterials on substrates and nanoparticles [49].

The surface chemistry of graphene has a great role in its cytotoxicity. By adhesion or bonding with cell receptors, bare graphene or GO can restrict the flow of nutrients, generate stress, and trigger apoptotic pathways, resulting in significant toxicity even at low concentrations. The use of biocompatible polymers or chemicals to functionalize graphene-based materials will dramatically minimize this cytotoxicity [50]. As a result, the nanomaterial’s surface properties, including shape, aggregation state, coatings, or functionalization, that may be present could play a part in their harmful impacts. The surface area/mass ratio and retention time of particles impact their toxicity; particles with a higher surface area and retention time interact more with cell membranes, allowing for increased absorption and transit into cells. As a result, the surface termination, size, and aggregation state of nanomaterials must be considered in any investigation, as they may have a major impact on the outcome of the biocompatibility tests.

### 2.3. Virus Detection

Currently, the COVID-19 pandemic represents a widespread health risk. Reducing COVID-19 transmission and infection in pre-symptomatic and asymptomatic persons needs ultrasensitive and early infection detection [51,52]. In order to return to normal routines in business, industry, schools, and universities, multiple strategies for reducing the risk that is associated with COVID-19 are required, including rapid, specific, easy, low-cost, and effective virus testing and monitoring [53,54]

In comparison with conventional virus detection techniques, the unique characteristics of graphene biosensors have been effectively utilized for the diagnosis of diseases [55,56,57]. Graphene oxide (GO) has played a vital role in the detection of different types of viruses, such as Rotavirus, Ebola virus, Influenza virus H3N2 hemagglutinin gene, AIDS (HIV), Hepatitis B virus (HBV), and Hepatitis C virus (HCV). The various assays and diagnostic tests that use the fluorescent biosensing mechanism are reviewed below [28,58]. Moreover, g-C3N4 was used to detect HBV via a simple, flexible, sensitive, and low-cost fluorescent biosensing device due to its high specific surface area and strong affinity [59].

#### 2.3.1. Rotavirus

Rotavirus is a gastrointestinal virus that causes infections and symptoms in newborns, children, and adults all over the world, making it a major cause of diarrhea [60]. Jung and coworkers developed a FRET-based GO biosensor for rapid Rotavirus detection. The AuNP–DNA–antibody formed a covalent bond with GO to prepare a sandwich structure, as shown in Figure 3 [61,62,63,64,65]. Virus binding onto the immobilized antibody took place via a specific antigen–antibody interaction. As a result of this, the fluorescence was quenched due to the interaction of the GO and AuNPs. The cross-reactivity of the biosensor was tested using Poliovirus and Variola virus. No emission of fluorescence was observed. However, a 15-fold increase in the fluorescence signal was observed with the addition of the Rotavirus. This biosensor achieved a linear dynamic range (10^3^ to 10^5^ pfu.mL^−1^). The unique fluorescence quenching property of GO sheets proves that GO can be applied in molecular diagnostic biosensors [64,65].

#### 2.3.2. Ebolavirus

The Ebola virus epidemic caused around 28,000 cases and 11,323 recorded deaths between 2014 and 2016. Ebola virus disease, widely known as a fatal disease, which can cause severe infection [66]. Wen and coworkers targeted the Ebola virus protein VP40, which interacts with the GO sheets, to develop a sensitive and selective fluorescent detection assay [51], using rolling circle amplification (RCA) (Figure 4). This fluorescent biosensor exhibits 1.4 pM LOD with a linear dynamic range from 30 fM to 3 nM. GO is negatively charged, and it directly interacts with the positively charged virus through π–π stacking. The GO sharp edges, was applied to lyse the virus envelope to start the amplification without the need for the addition of lysing agents or instrumentation [67]. The GO-based fluorescent biosensor offers an integrated lysing and sensing tool for the rapid detection of viruses [51].

#### 2.3.3. Influenza

Researchers are increasingly concerned about Influenza as a health threat. Influenza is a member of Orthomyxoviridae, consisting of a single-stranded RNA genome with four types (A, B, C, and D). Influenza type A consists of several surface antigens—for instance, hemagglutinin (H) and neuraminidase (N) [68]. Jeong et al. created a simple fluorometric platform using graphene oxide (GO) to detect Influenza. This platform relies on employing fluorescent DNA that is directly absorbed onto the GO via π–π stacking and hydrogen bonding (Figure 5), which results in the quenching of the FAM fluorescent molecule. The emission of the fluorescence of the FAM was increased upon addition of the Influenza RNA virus in a range from 37 to 9400 pg. This platform was able to detect as little as 3.8 pg Influenza RNA [38].

#### 2.3.4. HIV

A large number of patients over the world have died or became infected due to HIV/AIDS. As a result, a number of researchers have developed ultrasensitive diagnostic techniques for the early detection of HIV during the infection cycle [69,70]. Qaddare and Salimi developed a FRET-based biosensor to detect HIV, where carbon dots (CDs) were used as fluorescent NPs quenched with AuNPs, as shown in Figure 6A. This biosensor detected as little as 15 fM of the target oligonucleotides and had a dynamic range between 50.0 fM and 1.0 nM (Figure 6) [71].

HIV antibody detection was used as a sensitive and specific assay. Wu et al. designed a highly efficient FRET biosensor for the detection of anti-HIV-1 gp120 antibody via the interaction of GO with the peptide-functionalized UCNPs (Figure 7). The upconversion fluorescence intensity was increased linearly as the antibody concentration increased. The reported dynamic range was 5 to 150 nM of antibody concentration and D.L. was 2 nM [72].

Zhang et al. reported a new assay to detect HIV by labeling two partially complementary DNA probes (hairpin probe 1 (H1) and hairpin probe 2 (H2)), linked at one end with silver nanoclusters, as shown in Figure 8. They attached hairpin probes with silver nanoclusters (AgNCs) to GO. The developed system was exposed to the products of the hybridization chain reaction (HCR). The limit of detection for the reported assay was 1.18 nM [73,74].

Moreover, Zhang et al. developed a fluorescence biosensing technique to detect HIV-1 protease based on GO, using covalently bonded, fluorescently labeled HIV-1 protease target peptide molecules with GO. The fluorescence was increased upon the addition of HIV-1 protease as a result of the peptide cleavage and the release of the fluorescent peptide fragment away from the GO, as shown in Figure 9. This fluorescence-based detection assay was successfully detected as little as 18 ng/mL of HIV-1 protease [75].

#### 2.3.5. Hepatitis B Virus

Hepatitis is a liver inflammatory disease. It is often caused by a viral infection known as viral hepatitis that leads to 686,000 deaths per year [74]. HBV is a Hepadnaviridae virus that has its viral double-stranded DNA enclosed. Scientists have developed various biosensors with high sensitivity to be used in the early diagnosis of HBV [76].

Carbon nitrides, g-C3N4, a new family of carbon functional materials, have recently attracted the attention of researchers in the field of biosensing. This is a metal-free carbonaceous substance with a high N:C ratio and a structure that ranges from polymeric to graphitic, with heptazine or triazine rings serving as structural motifs [77]. The properties of g-C3N4, such as large specific area, optical properties, low-cost, biocompatibility, and the stability of the photoluminescence, make it ideal for the fluorescence sensing. Therefore, Xiao et al. developed an innovative fluorescent biosensor to detect the HBV gene based on the fluorescence quenching of g-C3N4 nanosheets (Figure 10). This unique design shows a great deal of potential for monitoring disease markers, especially HBV genes, because it offers a low-cost and quick response, with a 1.0 nM LOD [59].

#### 2.3.6. Hepatitis C Virus (HCV)

The Hepatitis C virus (HCV) is the most widespread cause of chronic liver disease, affecting 2–3% of the world population. In the past, early detection of HCV was difficult, especially for those living in underdeveloped nations. Jialong et al. employed reduced graphene oxide nanosheets (rGONS) with the hybridization chain reaction (HCR) amplification technique, to develop an ultrasensitive method for the detection of HCV RNA. The reported LOD was as low as 10 fM, which is substantially lower than the commonly used fluorescence approach based on GO [78] (Figure 11).

**Figure 11 biosensors-12-00460-f011:**
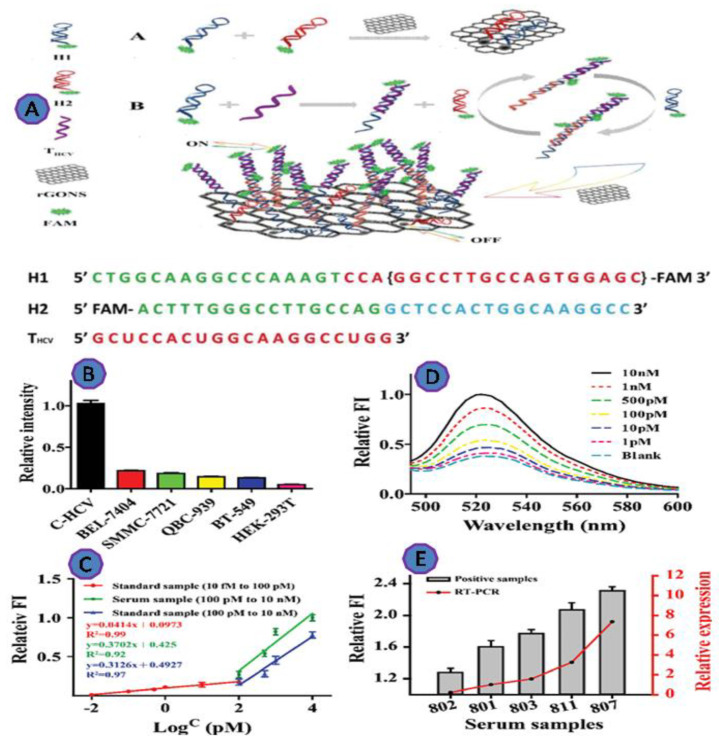
(**A**) Graphic depicting the proposed rGONS-HCR system for HCV detection; (**B**) THCV detection in various cell lysates; (**C**) fluorescence spectra of the rGONS-HCR in the presence of THCV at various concentrations in normal human serum; (**D**) the relative fluorescence intensity (FI) vs. THCV concentration; (**E**) THCV detection in clinical serum samples. Reprinted with permission from Ref. [79]. Copyright 2019, ROYAL SOCIETY OF CHEMISTRY.

### 2.4. Multiplexing Viruses Detection

One of the most important requirements for a biosensing platform is the simultaneous detection of multiple viruses infection (multiplexing) [80,81,82]. Different diseases can manifest with similar symptoms. Multiplex detection methodologies have become increasingly useful to detect a particular virus or viruses that cause an infection and their severity (quantitative) [51,82,83].

Chen et al. introduced a scalable, fluorescent biosensor for multiplex virus detection using graphene and 2D carbon materials. For the multiplexed analysis of virus genes, Chen et al. used an N, S co-doped GO platform for the detection of HBV as well as the detection of HIV. In this study, two complementary sequences were conjugated with various fluorescent dyes to the HPV and HIV virus genes, as shown in Figure 12A. This platform showed an improvement in the detection limit to as little as 2.4 nM (Figure 12) [79].

Another fluorescent multiplexing viruses detection method has been designed, through a hybrid system consisting of GO and nucleic acid-stabilized silver nanoclusters (AgNCs), for the detection of the HIV and HBV genes, as shown in Figure 13. This design can be used for the multiplexing detection due to the tunable fluorescence properties of AgNCs and the combination of AgNCs/DNA with GO. The conjugation system of ssDNA sequences and the DNA-stabilized AgNCs has demonstrated that adsorption to GO induces AgNC fluorescence quenching. The reported LODs for the genes of HIV and HBV were 1 nM and 0.5 nM, respectively [84].

## 3. Literature on 2D Fluorescent Biosensors for the detection of Viruses

Two-dimensional materials have been used extensively by various researchers to detect different viruses. Table 1 summarizes some of the developed 2D fluorescent biosensors for the detection of various viruses using various recognition elements, their mechanisms of operation, and their limits of detection.

### 3.1. Overcoming the Drawbacks in Carbon Nanomaterials

The performance of the fluorescent biosensors can be influenced by different characteristics, such as sensitivity, the limit of detection (LOD), repeatability, as well as reproducibility. The fluorescence transduction mechanism depends on the morphological parameters, and coordination surface chemistry can be used to examine the reliability and capability of sensors [49].

The drawbacks of stability and nonspecific absorption of GO-based DNA have attracted more attention due to the decreasing electron transfer efficiency [35]. Some works have shown that these problems can be overcome by using the heteroatom doping technique. Heteroatom doping is a characteristic technique was used to influence the electronic properties within the GO surface. The possibility of developing a new electronic structure on the sp^2^-hybridized carbon has gained a large amount of interest in terms of co-doping multiplex heteroatoms to enhance the activity of bare GO. For example, N, S co-doped GO with significant sulfur loading has been shown to efficiently break carbon material’s inertness, activate the sp^2^-hybridized carbon lattice, and permit electron transport from electron donor functional groups to GO sheets [82].

### 3.2. Detection Limit and Analysis Time

In short, the literature suggests that the detection limit is strongly determined by the sensitivity of the transducer, the mass transport kinetics and how fast the binding of the analyte with the recognition element [85]. Firstly, the sensitivity of the transducer can be determined by detecting the smallest concentration of the analyte that binds to the sensor’s surface, taking into consideration the differentiation of the signal-to-noise ratio that occurs due to the nonspecific binding. The mass transport kinetics and reaction of the analyte with the recognition element can influence the analysis time [86].

Improving the detection limit and shortening the analysis time can be achieved by selecting highly efficient recognition elements that bind tightly and rapidly to the analyte(s) in question. Furthermore, the recognition element should have high specificity to the analyte in question, to reduce the cross-reactivity, and the sensor surface should be blocked well, to reduce the non-specific binding and to reduce false positive signals. Another means for shortening the analysis time is to enhance the transfer of the analyte to the immobilized recognition element onto the sensor surface. This includes integrating the sensor surface with various electrokinetic techniques, ultrasound standing waves, optical trapping and concentration, and the usage of pre-concentration techniques prior to analysis.

### 3.3. Non-Specific Binding

Generally, non-specific binding of the analyte, can cause some limitations in the biosensing platforms. Reliable biosensor devices must have the ability to distinguish between the target analyte of interest and the other species present in the sample. The majority of the competing species can interact non-specifically with the sensing platform, which can cause a false positive signal. To overcome this challenge, some approaches have developed 2D material-based sensors that show more specific properties for the species in question. Graphene is one of these 2D materials that introduces specific binding, and the resulting fluorescence comes only from the binding species, and not from the interfering species [87]. Graphene is functionalized via either non-covalent or covalent binding. Most of the covalent methods present very complex and difficult conditions to control.

External parameters such as temperature, pH, salt content, etc. that might affect the interaction of the biomolecules with the surface of the graphene, as well as their unique characteristics, represent a challenge for improving the efficiency. In addition, the heterogeneity of the graphene has great significance for the performance of fluorescent biosensors based on different factors, such as size, thickness, and the number of functional groups on the surface [31].

One mean of improving the sensor’s specificity is to develop high-affinity and specific biosensing platforms via non-covalent attachment of the recognition elements to the surfaces of 2D carbon nanomaterials [88].

### 3.4. Cytotoxicity

The carbon element is intrinsically compatible with living systems, therefore, carbon-based nanomaterials are considered “safe”. Although graphene-based sensors have received a great deal of attention in health monitoring and biomedical applications. Numerous studies have investigated the graphene-based nanomaterials’ biocompatibility, water-solubility, toxicity, and potential environmental risks. As a result, the increased use of graphene in a variety of applications and industries will require different toxicological profiles in-vitro and in-vivo [64,89]. However, a variety of carbon nanomaterials, such as carbon nanotubes, nanodiamonds, and graphene oxide, have been examined in living cell lines and found to have little cytotoxicity [41]. The morphology, size, shape, purity, and surface chemistry have a powerful impact on the cytotoxicity, while the functionalization of graphene can adjust its interactions with the living cells [90]. Due to the lack of in-vivo applications, the cytotoxicity properties of graphene are still unknown. Several factors, including the lateral scale, oxygen content, and dose, need to be carefully researched [20,43,64].

A cytotoxicity evaluation of graphene using rat pheochromocytoma cells (PC12) observed 40% cell death using the MTT assay [91]. The cytotoxicity of GO on human lung carcinoma (A549) cells was observed with no cellular uptake, which resulted in decreased viability at higher concentrations. Moreover, the cytotoxicity of graphene depends on the quality of its dispersion [89]. Zhang et al. noticed an increase in the cytotoxicity of graphene when applied above 10 μg/mL. The study observed no pathological alterations with the use of distributed single-layered GO sheets of 10–800 nm in lateral size in Kunming mice, at a dose of either 1 mg/kg or 10 mg/kg [92]. It was noticed that the presence of 10% FBS in the cell media decreased the cytotoxicity of GO because of the direct physical damage of the plasma membrane caused by GO [93,94].

## 4. Future Outlook and Challenges

Great efforts have been devoted to designing new graphene-based FRET sensing systems for clinical testing, demonstrating inexpensive biosensing platforms with the eco-friendly, sustainable synthesis of graphene to solve the existing stability issues and efficiently enhance the electron transfer rate in enzymatic biosensors [95,96].

Graphene-like 2D nanomaterials such as graphene, GO, and rGO, with their functional reactive groups such as –OH, –COOH, –CO, and –C–O–C, are crucial for the immobilization of recognition elements. The strong quenching efficiency of GO and rGO in comparison with graphene offers exciting opportunities for the development of fluorescent biosensors with significant improved sensitivity and selectivity, and low limits of detection. The amphiphilic character of the GO facilitates the adsorption of diverse biomolecules on its surface due to its strong binding affinity with the biomolecules via pi–pi stacking. Several challenges still hinder the practical application of graphene-based smart materials—for example, the lack of techniques that can produce high-quality graphene materials at low-cost and in a more controllable and scalable fashion. It is necessary to investigate the toxicity of these materials—both in-vitro and in-vivo—via the functionalization of graphene-based smart materials. Because graphene assemblies are lightweight, flexible, and elastic, it is of great interest to develop graphene-based smart materials that can sensitively respond to multiple stimuli or are operational under harsh conditions (for example, high temperatures, high strength, and strongly corrosive media [97]). One of the most critical scientific challenges is to produce ultrasensitive fluorescent biosensors for rapid and specific detection at low-cost. Another challenge facing the graphene-based biosensors is the reproducibility and reliability of the materials production and processes [90].

To meet the ever-increasing demands of the fluorescent biosensors in nanomedicine applications, fluorescent biosensors must be able to distinguish the signals of the target analyte from the signal of the interferences. By applying different efficient recognition elements, they are expected to facilitate highly specific multiplexing viruses detection. Many of the current biorecognition elements are very reliable, while some of them have poor chemical stability, have a limited shelf-life, suffer from cross-reactivity, and are expensive, necessitating further advances in the synthesis and selection of emerging recognition elements to overcome the shortcomings. Multiplexing detection technologies are considered one of the main areas of future development.

Enhancing the signal-to-noise ratio, which is normally calculated by the receptor–nanomaterial–transducer interface, is another significant factor to consider. More research into the mechanisms of interactions between the DNA probes or modified DNA probes and the graphene-based transducer is required to provide more reliable and accurate measurements. The full potential of currently researched biorecognition elements is limited due to the lack of reproducibility and reusability of the biosensing platforms performance characteristics in the literature.

Despite the massive increase in the use of 2D carbon materials in point-of-care diagnostics, future studies considering their toxicology and biocompatibility are still needed.

To simplify healthcare and medicine in the future, a massive increase in the use of 2D carbon materials in point-of-care diagnostics studies will be necessary, such as touchscreen devices for epidermal electronic devices [98]. Huang et al. developed a cost-effective, fast, accurate, and specific microfluidic chip for point-of-care testing that allows for the simultaneous identification of various clinical pneumonia-related bacteria in 1.45 μL reactions, without cross-contamination, in 45 min [99]. Pang et al. designed a novel, rapid, and sensitive self-priming PDMS/paper hybrid (SPH) microfluidic chip for the detection of specific multiplex foodborne pathogens via mixed-dye loading. This device can detect eight pathogens simultaneously by increasing the number of reaction chambers in the SPH chip, demonstrating its significant potential for on-site food safety screening and wider application [100].

Point-of-care (POC) devices enable multiplex, portable, rapid, and economically affordable instrumentation, with respect to traditional culture or PCR-based assays. Such instrumentation can be further miniaturized into point-of-care (POC) devices. All these features can be effectively utilized in the detection of pathogenic microorganisms [29,101]. Using a smartphone translation tool, point-of-care (POC) devices look to be a promising step for physicians in lab analysis. Researchers are looking at new smartphone-connected POCT devices to improve data sensitivity, with additional advantages such as privacy and theft protection, medical recommendations, and time- and cost-intensive laboratory processing without sample pretreatment. They should be portable and user-friendly while still delivering appropriate analytical results and clinical relevance [102]. The sensitivity, low detection limit, and selectivity of the point-of-care (POC) devices must be improved by utilizing a sensing prototype based on a smart optoelectronic nanosystems, using precise diagnostics without interferents or loss to choose a real sample source. These procedures are used to validate and scale up the sensor for clinical use [103]. Due to the present COVID-19 pandemic, a reliable, rapid, cost-effective, and simple testing technique is urgently required. One improvement could be integration with lab-on-chip to clean up the sample and pre-concentrate the analyte of interest. The application of graphene in LOC devices has greatly increased their sensing capabilities [104].

As a result, graphene-based LOC devices can provide a robust microenvironment for advanced detection techniques at a low-cost [105]. Xiang et al. developed a microscale fluorescence-based colorimetric sensor employing a graphene nanoprobe for multiplexed DNA analysis [106]. A highly flexible microfluidic tactile sensor based on a GO nanosuspension was reported by Kenry et al. [107]. A microfluidic chip integrated with a novel GO-based Forster resonance energy transfer biosensor was fabricated by Cao et al. to detect cancer cells in-situ [96].

Graphene has great mechanical strength and also has the feature of flexibility. Due to its remarkable mechanical strength, graphene is a useful material in the production of wearable LOC devices. Graphene also has a large specific surface area, which helps surfaces to load the necessary ligand in the LOC devices for single-molecule detection. Furthermore, graphene has tunable optical characteristics, which are highly useful for optical readouts in fluorescent LOCs to detect a wide variety of viruses, ranging from Influenza virus to the present SARS-CoV-2 virus [105]. For the detection of H1N1, Joshi et al. fabricated a stable, reproducible, and green device based on rGO flakes [108]. For the detection of avian Influenza, Roberts et al. developed a functionalized graphene-based field-effect transistor (GFET) with an LOD of 10 fM [109]. For the detection of Ebola virus, Jin et al. obtained an LOD of 2.4 pg/mL based on the response of rGO with GFET as a function of the Dirac voltage [110]. For the detection of SARS-CoV-2, Simone et al. [111] theoretically studied the possibility of using a graphene-based optical sensor via employing Raman scattering, generating results within 1 min and having an LOD of 1.68 × 10 −22 μg mL^−1^. Torrente-Rodríguez et al. [53] designed a highly selective, unique, ultrasensitive, multiplexed, and rapid graphene-based immunosensor and telemedicine system for the detection of SARS-CoV-2 in blood and saliva samples.

## 5. Conclusions

The excellent properties of 2D carbon nanomaterials, such as high sensitivity and specificity, ease of operation, and fast detection, make them suitable to detect viruses, e.g., Influenza, Hepatitis B, HIV, and Rotavirus, using fluorescent biosensing techniques. This review focused on the advantages of 2D nanomaterial-based biosensors. Despite the developments and advancements in laboratory-based virus detection, practical implementations face challenges in terms of in-field applications. Additional excellent 2D carbon fluorescent biosensors will be proposed in the future, and new solutions will arise from collaborative efforts in the field of nanomedicine. As a result, further research into the feasibility and broad practical applicability of fluorescent biosensors for multiplexing detection is needed to augment the detection efficiency. The challenges for the researchers in this domain are (a) the need to improve efficiency through production and biofunctionalization, (b) the improvement of the systems’ robustness, (c) the application of biosensors in implantable systems, and (d) advancements in large-scale manufacturing strategies with small-sized devices.

In summary, the development of fluorescent biosensors based on graphene nanomaterials represents a major turning point in the medical field, which could improve the quality of life of many patients.

## Figures and Tables

**Figure 1 biosensors-12-00460-f001:**
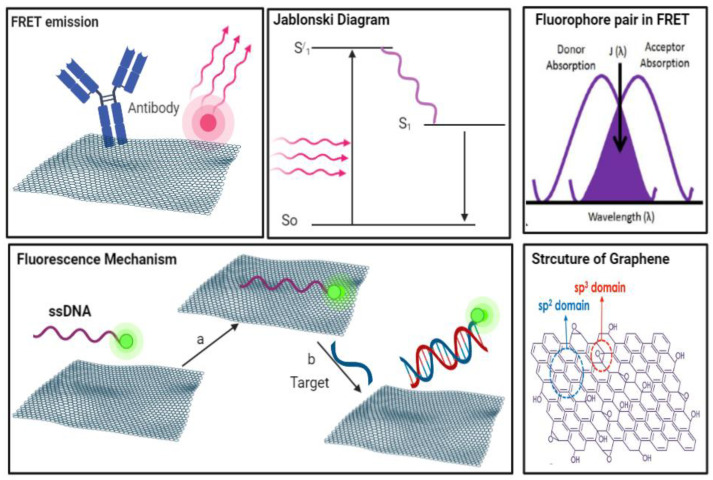
Illustration of the FRET emission mechanism on graphene oxide related to Jablonski diagram.

**Figure 2 biosensors-12-00460-f002:**
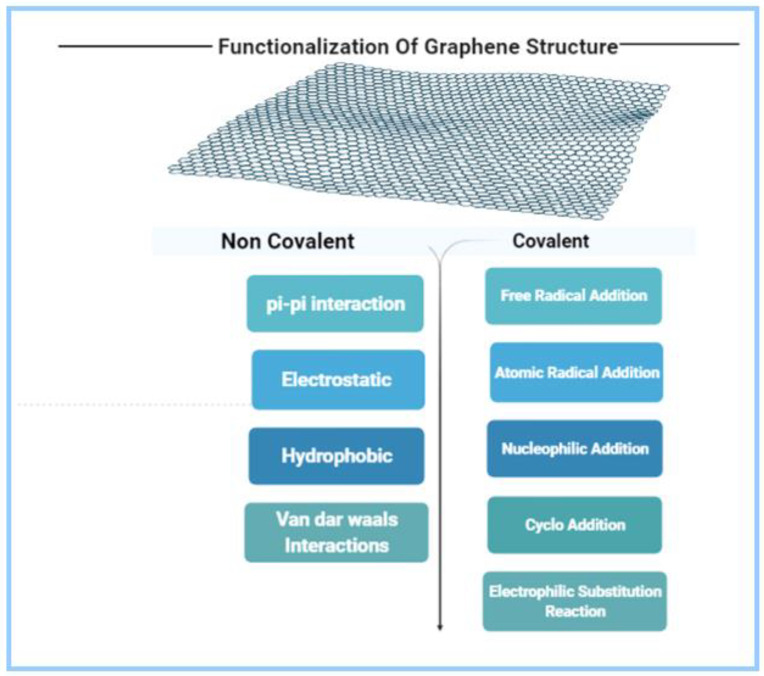
Different categories of functionalization of graphene oxide.

**Figure 3 biosensors-12-00460-f003:**
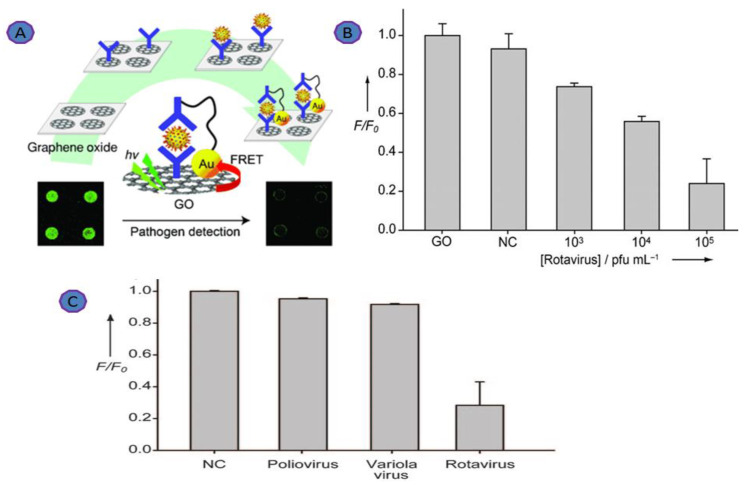
(**A**) Illustration of the GO-based immuno-biosensor; (**B**) dynamic range of the GO-based sensor; (**C**) cross-reactivity of the GO-based immunosensor was tested with Poliovirus and Variola virus. Reprinted with permission from Ref. [65]. Copyright 2010, Wiley-VCH.

**Figure 4 biosensors-12-00460-f004:**
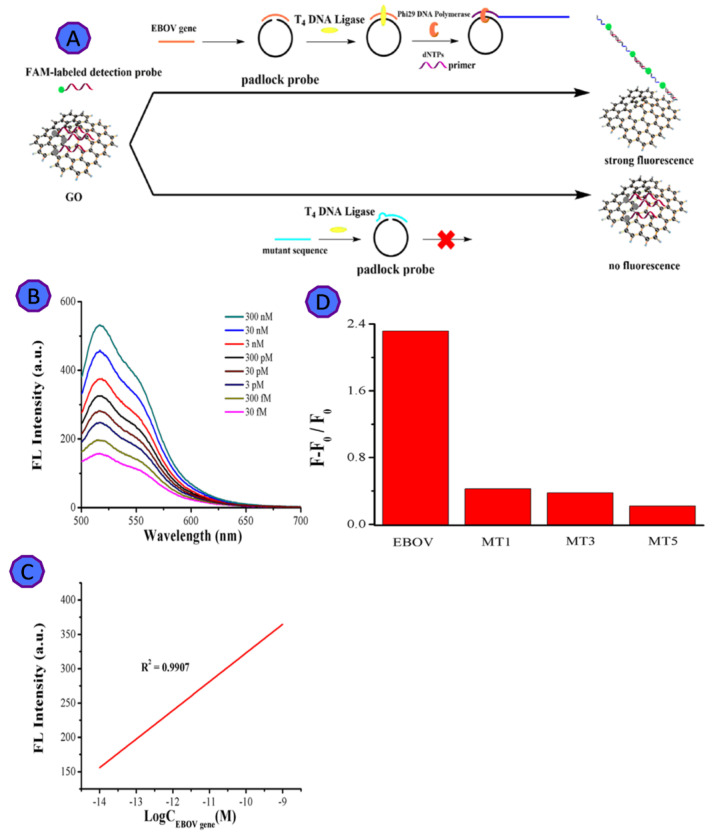
(**A**) The graphical representation shows the GO-assisted amplified biosensor for EBOV detection; (**B**) fluorescence increase as a function of virus increase; (**C**) the relationship between fluorescent intensity and the concentration of EBOV gene; (**D**) the relationship between the (F − F_0_)/F_0_ of GO and EBOV gene, with single-base mismatched sequence (MT1), three-base mismatched sequence (MT3), and five-base mismatched sequence (MT5). Reprinted with permission from Ref. [67]. Copyright 2022, Elsevier B.V.

**Figure 5 biosensors-12-00460-f005:**
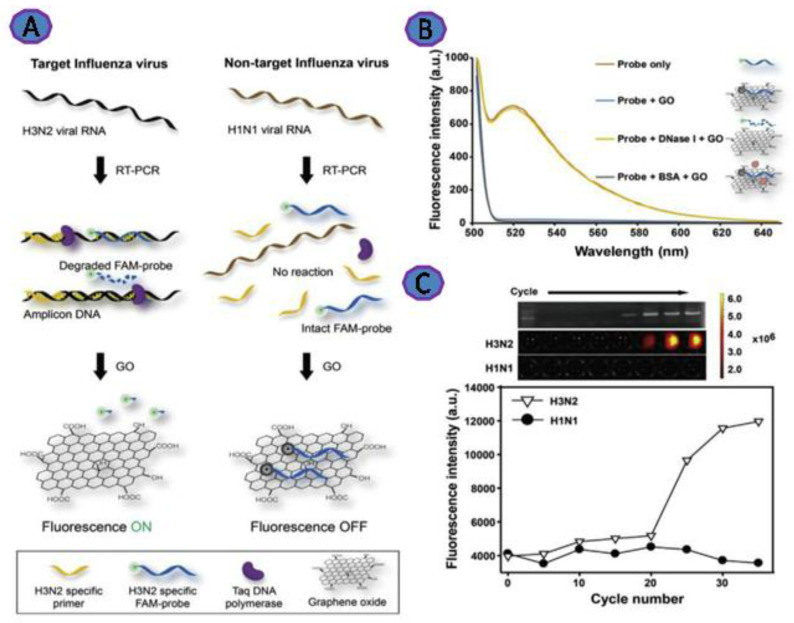
(**A**) Representation of fluorescence detection of Influenza virus RNA via GO and Taq polymerase’s 5′ to 3′ exonuclease activity during RT-PCR; (**B**) representation of the relationship between the fluorescence emission spectra of the FAM-DNA probe and the DNase I after GO incubation (λex = 485 nm); (**C**) with each PCR cycle, the fluorescence of the FAM-DNA probe increased. RT-PCR was performed on H3N2 (target) and H1N1 (non-target) viral RNAs using H3N2 hemagglutinin gene-specific primers and the FAM-DNA probe. Reprinted with permission from Ref. [38]. Copyright 2022, Elsevier B.V.

**Figure 6 biosensors-12-00460-f006:**
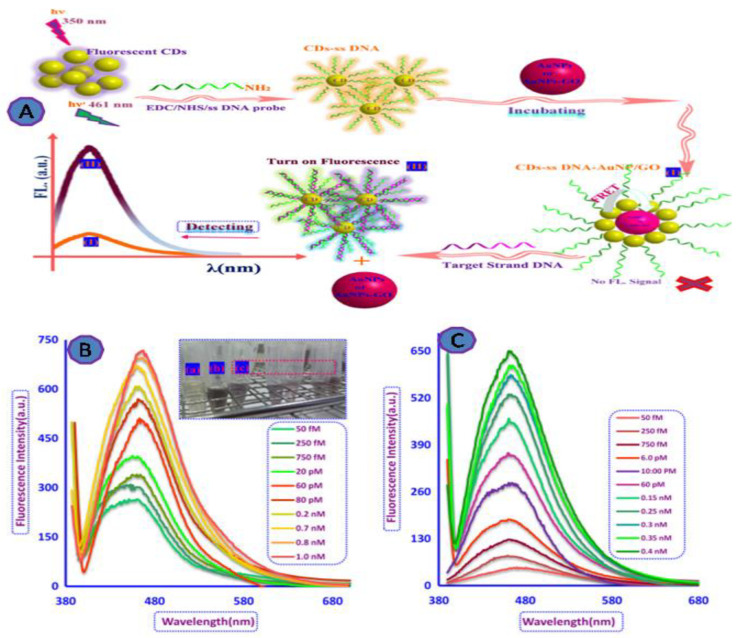
GO-fluorescent biosensor for HIV viral detection. (**A**) The scheme of the FRET-based detection system; (**B**) fluorescence recovery spectra of the capture probe ssDNA-CD/AuNP system in the presence of various amounts of target DNA at λ 350 nm excitation and 0.22 nM AuNPs as a quencher; (**C**) the fluorescence recovery spectra of the ssDNA-CD-AuNP/GO system in the presence of different concentrations of target DNA at excitation wavelength of 350 nm and 5 μg/mL of GO/AuNPs as quencher. Reprinted with permission from Ref. [71]. Copyright 2017, Elsevier.

**Figure 7 biosensors-12-00460-f007:**
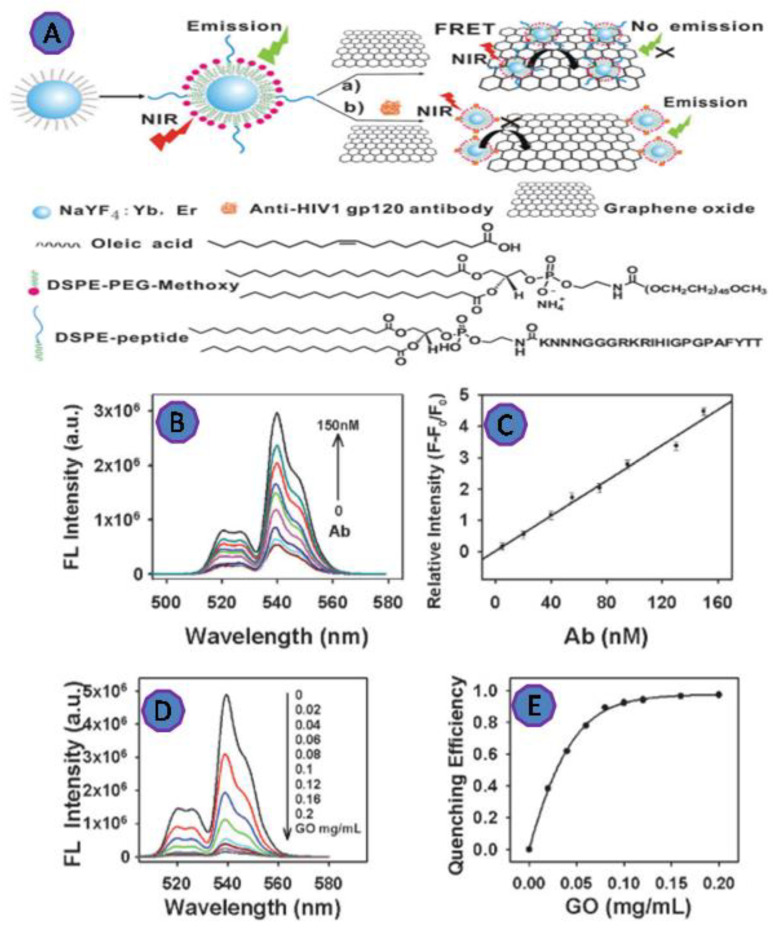
(**A**) Scheme of the upconversion FRET-based biosensor for detecting anti-HIV-1 gp120 antibody; (**B**) the upconversion fluorescence emission of the biosensor; (**C**) the calibration curve of the UCNPs showed a linear response with the various antibody concentrations in the range of 5–150 nM; (**D**) the fluorescence spectra of the peptide-functionalized UCNPs after incubation with varying concentrations of GO; (**E**) fluorescence quenching efficiency against GO concentration. Reprinted with permission from Ref. [72]. Copyright 2022, ROYAL SOCIETY OF CHEMISTRY.

**Figure 8 biosensors-12-00460-f008:**
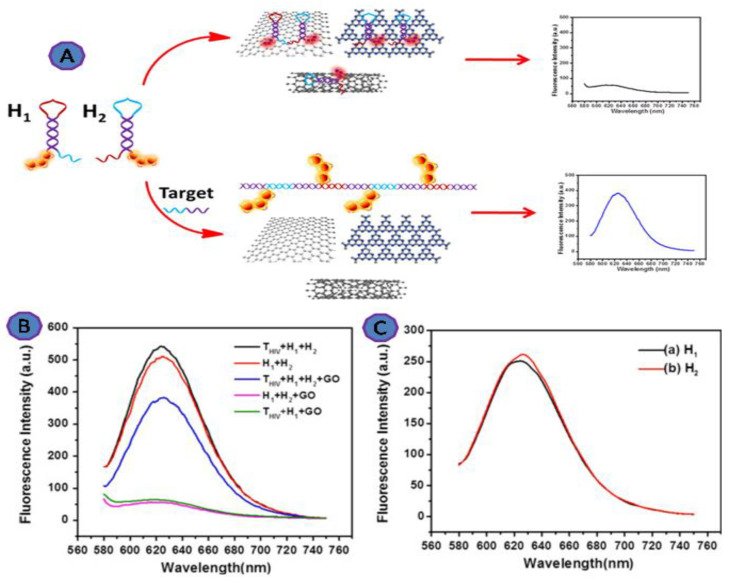
(**A**) Scheme showing graphene oxide (GO)-based HCR for DNA detection and the fluorescence response of the system under different conditions; (**B**) THIV + H1 + H2; H1 + H2; THIV + H1 + H2 + GO; H1 + H2 + GO; THIV + H1 + GO; (**C**) H1; H2. [H1] = 100 nM, [H2] = 100 nM, [THIV] = 100 nM. Reprinted with permission from Ref. [73]. Copyright 2017, Elsevier.

**Figure 9 biosensors-12-00460-f009:**
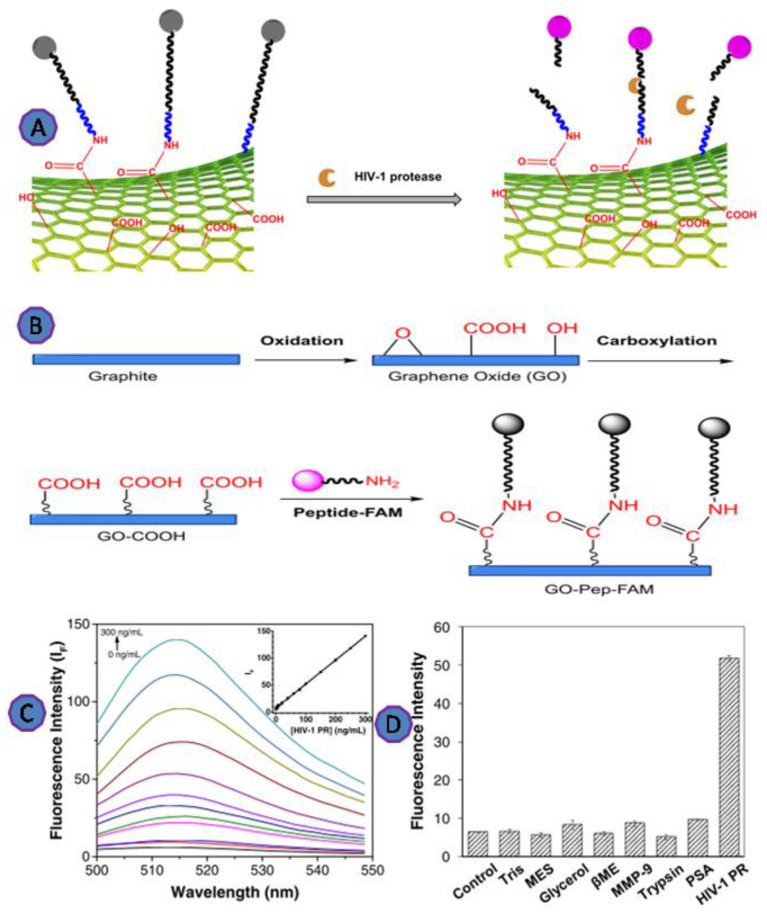
(**A**) Scheme depicting the principles of detecting HIV-1 protease; (**B**) scheme depicting the production of an HIV-1 protease sensor using graphite powder; (**C**) fluorescence spectra of the GO-Pep-FAM sensor in the presence of HIV-1 protease at various doses; (**D**) plot of fluorescence intensity versus various species, demonstrating the selectivity of the GO-Pep-FAM sensor. Reprinted with permission from Ref. [75]. Copyright 2018, Springer Nature.

**Figure 10 biosensors-12-00460-f010:**
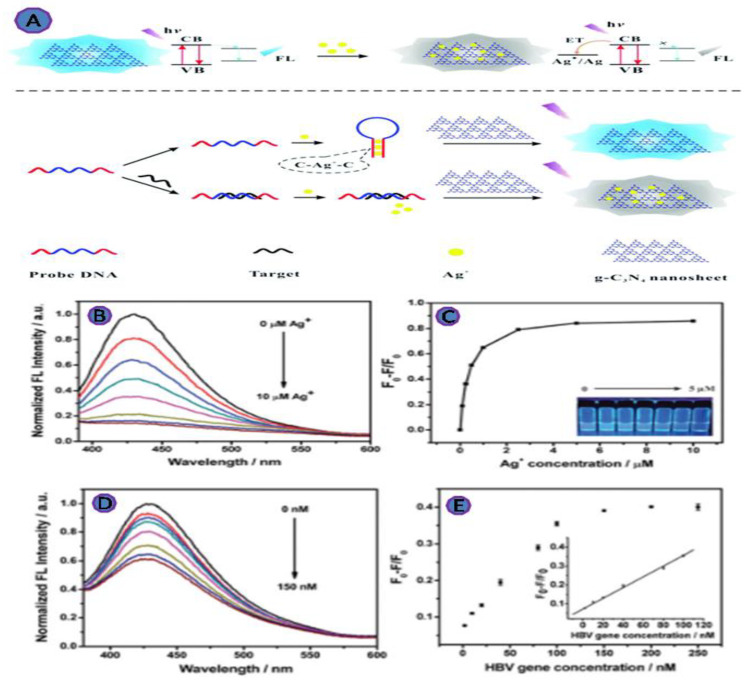
(**A**) A diagram of the method for detecting pathogenic DNA based on metal ion binding-induced fluorescence quenching of graphitic carbon nitride nanosheets; (**B**) fluorescence spectra of g-C3N4 nanosheets (1.0 gmL^−1^) in the presence of various silver ion concentrations; (**C**) the fluorescence intensity of g-C3N4 nanosheets was measured in the presence of various numbers of silver ions; (**D**) fluorescence spectra of g-C3N4 nanosheets in the presence of different HBV gene concentrations; (**E**) the relative fluorescence intensity of g-C3N4 nanosheets in response to various HBV gene concentrations. Reprinted with permission from Ref. [60]. Copyright 2017, ROYAL SOCIETY OF CHEMISTRY.

**Figure 12 biosensors-12-00460-f012:**
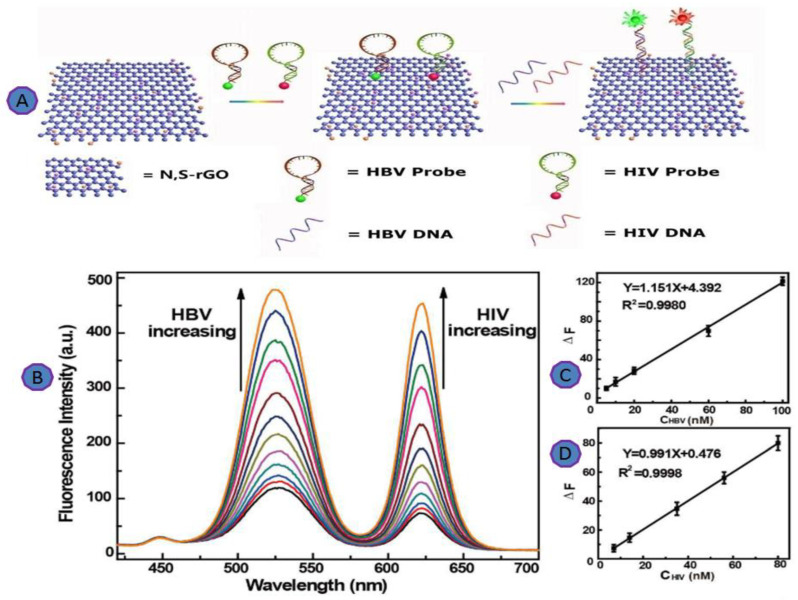
(**A**) Schematic illustration of N, S-rGO for multiplexing detection of HBV and HIV; (**B**) fluorescence spectra of N, S-rGO; (**C**) linear relationship for the detection of HBV DNA; (**D**) linear relationship for the detection of HIV DNA. Reprinted with permission from Ref. [77]. Copyright 2016, American Chemical Society.

**Figure 13 biosensors-12-00460-f013:**
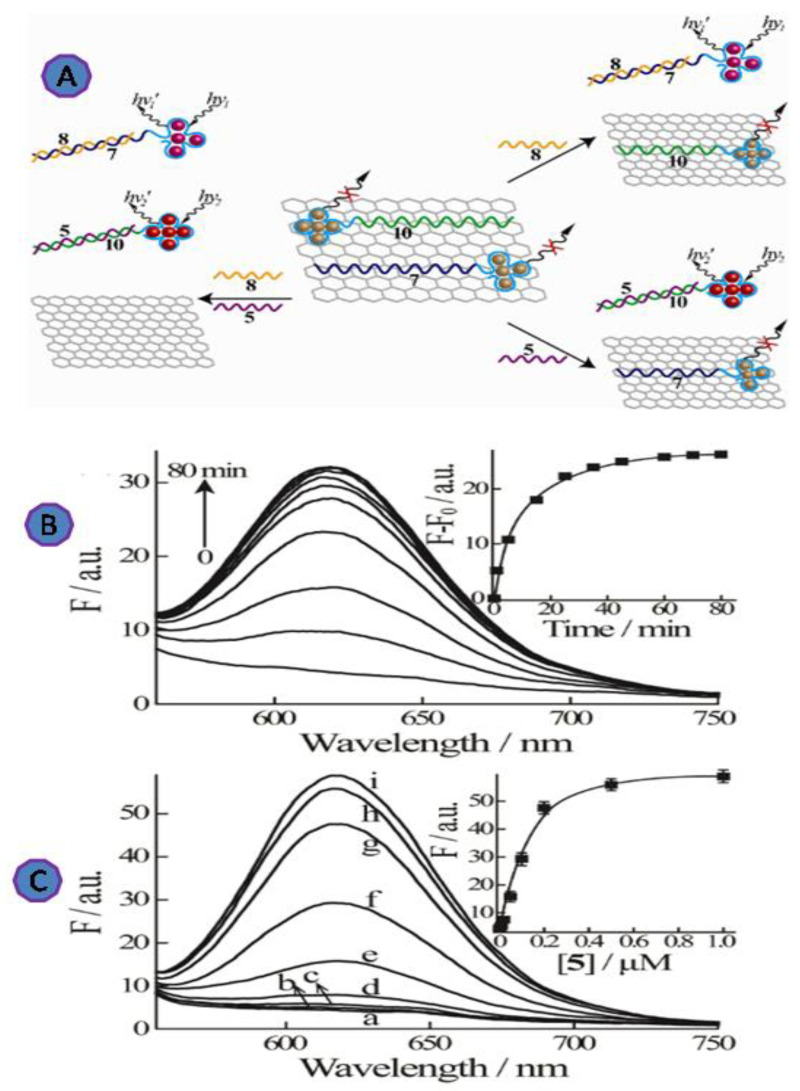
(**A**) Schematic diagram of the multiplexed detection platform for the HBV gene (5) and the HIV gene using near-infrared and red-emitting AgNC probes (8); (**B**) time-dependent fluorescence spectra of the red-emitting AgNCs upon challenging the (6)-AgNCs/GO with the HBV target gene (5), 100 nM. Inset: Time-dependent fluorescence changes at λ = 616 nm upon analyzing (5) by the (6)-AgNCs/GO. (**C**) Fluorescence spectra of the AgNCs upon analyzing different concentrations of the HBV gene (5) by the (6)-AgNCs/GO system. (a) 0, (b) 2 nM, (c)10 nM, (d) 20 nM, (e) 50 nM, (f) 100 nM, (g) 200 nM, (h) 500 nM, and (i) 1000 nM. Fluorescence spectra were recorded after a fixed time-interval of 80 min. Inset: Calibration curve corresponding to the luminescence of the related AgNCs (at λ = 616 nm) in the presence of different concentrations of (5). Reprinted with permission from Ref. [84]. Copyright 2013, American Chemical Society.

**Table 1 biosensors-12-00460-t001:** List of the 2D fluorescent biosensors available for virus detection, types of employed recognition element, dynamic range, and limits of detection (LOD).

2D Carbon Structure	Virus Sensing	Recognition Element	Type of Interaction	Dynamic Range	Limit of Detection	REF
GO	Rotavirus	Antibodies	Carbodiimide-assisted amidation reaction	10^3^ to 10^5^ pfu mL^−1^	10^5^ pfu mL^−1^	[64,65]
GO	Ebola virus gene Influenza	dsDNA	π–π stacking interaction	30 fM–3 nM	1.4 Pm	[67]
GO	Influenza virus H3N2 hemagglutinin gene	RNA	π–π stacking interaction and hydrogen bonding	37–9400 pg	3.8 pg	[38]
GO	HIV-1 gene	dsDNA	carbodiimide-assisted amidation reaction	50.0 fM–1.0 nM	15 fM	[71]
GO	HIV	Antibodies	π–π interaction	5–150 nM	2 nM	[72]
GO	HIV	dsDNA	π–π interaction	-	1.18 nM	[73,74]
GO	HIV	Enzyme	π–π stacking and/or electrostatic interaction between	5–300 ng/mL	109 pM	[75]
rGO	HCV	dsDNA	π–π interaction		10 fM	[78]
g-C3N4	HBV gene	DNA	π–π stacking	2–100 nM	1.0 nM	[59]
rGO	HIV	Aptamer	π–π interaction	-	3.0 nM	[79]
rGO	HBV	Aptamer	π–π interaction	-	2.4 nM	[79]
GO	HBV	ssDNA	π–π stacking	-	0.5 nM	[84]
GO	HIV	ssDNA	π–π stacking		1 nm	[84]

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
