# Peer review of "Fluorescent Biosensors for the Detection of Viruses Using Graphene and Two-Dimensional Carbon Nanomaterials"

_biosensors, 2022, doi:10.3390/bios12070460_

Round 1
Reviewer 1 Report
This paper reviews graphene and two-dimensional carbon materials-based Fluorescent bio- 13 sensors are covered between 2010 to 2021, for the detection of different human viruses. This review 14 specifically focuses on the new developments in the graphene and two-dimensional carbon nonmaterial for fluorescent biosensing. In my opinion, this review paper could be accepted for publication on Biosensor the after the revisions as follows.
*To insight for reader on search engines, add two more keyword.
* Pay attentions to the format as introduction and second heading are out of format. * Explain the pros and cons of graphene biosensor over others and also explain the role of aptamer in graphene fluorescence sensing.
* There is also need to provide the graphical description.
* Give a brief about graphene and two-dimensional carbon materials-based Fluorescent biosensors history, toxicology and all expected sensing mechanism.
* The author related to FRET, also need to explain in abstract and include in your objective.
* There are many mistakes and misimpressions in the manuscript such as line#260 should be humanity. The level of English throughout the manuscript is need to improve. It contains too many errors regarding format such as font size, case. Go through the whole manuscript to remove ambiguity and flaws regarding citation and references style and format.
* The layout of the graph is very disorderly and some of the Figures are not clearly. The references missing and copyright licensing is not appeared.
*The prospective and the challenges and the possible developments orientations should be stated in the last part.
*The big problem of the review is lack of personal opinions in the whole manuscript, and therefore, it is just like the summarization of the recent progress of sensors

Author Response
Dear Colleague,
Thanks for all of your efforts and time to improve our review. We took all of your comments in the reviewing process. We really appreciate all of your efforts.
Reply to reviewer 1
- To insight for reader on search engines, add two more keyword.
Thanks for the reviwer. We included: Limit of detection , Recognition element
- Pay attentions to the format as introduction and second heading are out of format
Thanks for the reviwer for his comment to improve our review. Sure done
- Explain the pros and cons of graphene biosensor over others also explain the role of aptamer in graphene fluorescence sensing.
Thanks for the reviwer. It has included.
- There is also need to provide the graphical description. Graphic abstract
The graphic abstract illustrates the mechanism of the developed fluorescent biosensors for viruses detection.
- Give a brief about graphene and two-dimensional carbon materials-based Fluorescent biosensors history, toxicology and all expected sensing mechanism.
Thanks for the reviwer. We included it
- The author related to FRET, also need to explain in abstract and include in your objective.
Thanks for the reviwer. Included
- There are many mistakes and misimpressions in the manuscript such as line#260 should be humanity. The level of English throughout the manuscript is need to improve. It contains too many errors regarding format such as font size, case. Go through the whole manuscript to remove ambiguity and flaws regarding citation and references style and format.
Thanks for the reviwer for his concern to improve the manuscript. We did proof reading it a number of times.
- The layout of the graph is very disorderly and some of the Figures are not clearly. The references missing and copyright licensing is not appeared.
Thanks for the reviwer for his comment. We arranged them now in better way. Figure 1 and figure 2 drawn by the first author. So no need for the copyright permission.
- The prospective and the challenges and the possible developments orientations should be stated in the last part.
Thanks for the reviwer comment to imorve the review. We included many parts.

Reviewer 2 Report
- The section of cytotoxicity needs a significant elaboration.
- The proposed alternative approaches to address challenges needs more elaboration.
- Prospects of point-of-care should be discussed in details. https://iopscience.iop.org/article/10.1149/2754-2726/ac5ac6/meta
- Abstract needs more scientific information to proposed the need to graphene and C-nano systems
- A comparative preference of graphene and C-nano systems should be discussed in introduction as well.
- A careful proof-reading is also required.
Overall, this review article is of high significance but needs a major revision.
Author Response
Dear Colleague,
Thanks for all of your efforts and time to improve our review. We took all of your comments in the reviewing process. We really appreciate all of your efforts.
Reply to reviewer 2
- The section of cytotoxicity needs a significant elaboration.
Thanks for your comment to improve the review. We included more information about this.
- The proposed alternative approaches to address challenges needs more elaboration.
Thanks for your comment. We improved it .
- Prospects of point-of-care should be discussed details. https://iopscience.iop.org/article/10.1149/2754-2726/ac5ac6/meta
Thanks for your comment. We included some information from literature. However it should be noted that not much work has been reported with graphene-based LOC.
- Abstract needs more scientific information to proposed the need to graphene and C-nano systems
Thanks for the comment to improve the review. We included some information related to this.
- A comparative preference of graphene and C-nano systems should be discussed in introduction as well.
Thanks for the nice comment to improve the review. We included some information about this.
- A careful proof-reading is also required.
Thanks for your comment to improve the review. We did extensive proof reading for the review.

Round 2
Reviewer 1 Report
The author greatly improved the revised manuscript. It can be accepted
Author Response
thanks alot for your valuable comments to improve our manuscript
Reviewer 2 Report
Authors seems rigid and not completely address the points in the revision.
For example, in the response, not so much work is reported on graphene for POC, I disagree.
besides, this is a review - always articulated for broad readership. Authors should share their viewpoint as this journal is reputed and has no word limitation.
If there no futuristic POC approach then this article has no novelty.
https://www.mdpi.com/2079-6374/11/11/433
https://www.mdpi.com/2079-6374/11/10/359
Although, authors efforts to perform revision is appreciated.
Author Response
Plz see attached file